

# Comparison of iCOR and Rayleigh atmospheric correction methods on Sentinel-3 OLCI images for a shallow eutrophic reservoir

Stefanos Katsoulis-Dimitriou[1], Marios Lefkaditis[1], Sotirios Barmpagiannakos[1], Konstantinos A. Kormas[1] and Aris Kyparissis[2]

[1] Department of Agriculture Ichthyology & Aquatic Environment, University of Thessaly, Volos, Magnesia, Greece

[2] Department of Agriculture Crop Production and Rural Environment, University of Thessaly, Volos, Magnesia, Greece

Corresponding author
Aris Kyparissis, akypar@uth.gr

## ABSTRACT

Remote sensing of inland waters is challenging, but also important, due to the need to monitor the ever-increasing harmful algal blooms (HABs), which have serious effects on water quality. The Ocean and Land Color Instrument (OLCI) of the Sentinel-3 satellites program is capable of providing images for the monitoring of such waters. Atmospheric correction is a necessary process in order to retrieve the desired surface-leaving radiance signal and several atmospheric correction methods have been developed through the years. However, many of these correction methods require programming language skills, or function as commercial software plugins, limiting their possibility of use by end users. Accordingly, in this study, the free SNAP software provided by the European Space Agency (ESA) was used to evaluate the possible differences between a partial atmospheric correction method accounting for Rayleigh scattering and a full atmospheric correction method (iCOR), applied on Sentinel-3 OLCI images of a shallow, highly eutrophic water reservoir. For the complete evaluation of the two methods, in addition to the comparison of the band reflectance values, chlorophyll (CHL) and cyanobacteria (CI) indices were also calculated and their values were intercompared. The results showed, that although the absolute values between the two correction methods did not coincide, there was a very good correlation between the two methods for both bands' reflectance ($r > 0.73$) and the CHL and CI indices values ($r > 0.95$). Therefore, since iCOR correction image processing time is 25 times longer than Rayleigh correction, it is proposed that the Rayleigh partial correction method may be alternatively used for seasonal water monitoring, especially in cases of long time-series, enhancing time and resources use efficiency. Further comparisons of the two methods in other inland water bodies and evaluation with *in situ* chlorophyll and cyanobacteria measurements will enhance the applicability of the methodology.

## INTRODUCTION

### Harmful algal blooms (HABs)

Recurrent blooms of harmful algae and cyanobacteria (HABs) in coastal and inland water systems are a major concern for environmental and public health authorities worldwide. HABs are associated with eutrophication and in particular with phosphorus and nitrogen loading due to runoff from rural areas (*Anderson et al., 2008*; *Heisler et al., 2008*; *Mazard et al., 2016*; *O'Neil et al., 2012*) and also with climate change and $CO_2$ concentration and temperature increase (*Glibert, 2020*; *Gobler, 2020*; *Griffith & Gobler, 2020*; *O'Neil et al., 2012*; *Raven, Gobler & Hansen, 2020*). Other environmental factors are high pH and light (*Bartosh & Banks, 2007*; *Ni et al., 2012*).

The effects of these blooms may concern changes in taste and smell of water supply sources, development of a thick crust on the surface of the lake and lack of water clarity (*Chorus & Bartram, 1999*). Additionally, from a biological perspective, possible toxic effects on other algae, invertebrates and fish and anoxic conditions that alter the structure of benthic macro-invertebrates may appear (*Havens, 2008*). Finally, the toxic secondary metabolites produced by blue–green algae may cause serious health problems in mammals and wildlife, because they affect the endocrine, dermal and nervous systems (*Carmichael, 2001*).

Effective HAB monitoring remains a challenge, as *in-situ* samplings are time-intensive and costly and provide information only at discrete locations in space and time (*Pokrzywinski et al., 2022*). As a common practice, in small inland waters, samples are taken either in the middle of the waterbodies by boat or near the shore and then analyzed in the laboratory. Therefore, remote sensing techniques, with their advantages on spatial and temporal resolution, are increasingly used to record and monitor HABs in inland waters, as seen in several recent studies (*Cicerelli, Galo & Roig, 2017*; *Duan et al., 2022*; *Ho et al., 2017*; *Kislik et al., 2022*; *Kudela et al., 2015*; *Ogashawara et al., 2013*; *Pompêo et al., 2021*).

### Sentinel-3 OLCI

The advancements in satellite remote sensing during the last decades has led to its significant contribution in numerous environmental applications. The Sentinel-3 satellites are a mission organized by the European Space Agency (ESA) and the European Meteorological Satellite Exploitation Agency (EUMETSAT) under the Copernicus program, formerly known as the Global Monitoring for Environment and Security (GMES). The Sentinel-3 mission comprises two similar satellites A and B, with launch dates February 16, 2016 and April 25, 2018 respectively. Both satellites are solar-synchronous with a polar orbit, operating at an average altitude of 815 km and with an inclination of 98.6° (*Yang, Zhang & Wang, 2019*). Among others, they carry an Ocean and Land Color Instrument (OLCI), which covers a spectral range from 400 to 1020 nm (21 spectral bands), with 300 m spatial resolution and approximately daily revisit cycle.

Ocean Color images have been available since 1978 thanks to the Coastal Zone Color Scanner (CZCS) mission. From 2002 to 2012, the Medium Resolution Imaging Spectrometer (MERIS) on ESA's ENVISAT platform provided unprecedented monitoring

capability for coastal and inland water systems (*Kravitz et al., 2020*). The OLCI Instrument onboard the Sentinel-3 satellites is based on the mechanical and imaging design of ENVISAT MERIS and may be considered as MERIS heritage for ocean and land color monitoring. It includes six more spectral bands than MERIS (21 to 15), which are centered at 400 and 674 nm (water constituents retrieval improvement), 761, 764, and 768 nm ($O_2$ gas absorption correction improvements), and 1020 nm (atmospheric correction improvement) (*Mograne et al., 2019*). According to *Shi et al. (2019)*, analysing the latest research for cyanobacterial bloom remote sensing in inland waters, MERIS was the optimal past sensor for providing detailed cyanobacterial bloom information products due to its radiometric, spectral, temporal, and spatial resolutions and OLCI has the same suitability as MERIS in deriving cyanobacterial bloom information in inland waters.

## Atmospheric correction

Satellite sensors measure the top-of-atmosphere (TOA) signal from the surface-atmosphere system in visible and near-infrared parts of the spectrum. The atmospheric path radiance received by a sensor at the TOA can be mainly decomposed into Rayleigh and aerosol scattering (*Feng et al., 2018*), with the Rayleigh-scattering radiance being the most dominant component of the TOA signal (*Shanmugam, Shanmugam & He, 2019*). The process of removing the atmospheric path signal from the TOA signal in order to retrieve the desired surface-leaving radiance signal is referred to as atmospheric correction (*Gordon, 1997*). It is worth noting, that after five years of Sentinel-3 operation, ESA is still providing only limited atmospherically corrected (Level 2) data. So, for acquiring information about the environment (for example monitoring of cyanobacterial blooms) from the satellite images, atmospheric correction is necessary and several atmospheric correction algorithms have been developed through the years.

## iCOR & Rayleigh atmospheric correction methods

iCOR is a free open-source atmospheric correction software (*Ibrahim et al., 2018*; *Nurgiantoro et al., 2019*) that can be used as an ESA Sentinel Application Platform (SNAP) plug-in for processing Landsat-8 OLI, Sentinel-2 MSI and Sentinel-3 OLCI images. iCOR runs with minimum user interaction, derives the required input parameters from the image and is designed to be applicable to inland waters, coastal waters and land.

The iCOR workflow includes four steps (*König, Hieronymi & Oppelt, 2019*; *Stefan et al., 2018*): (1) classification of land/water pixels (2) AOT retrieval over land following the approach of *Guanter, González-Sanpedro & Moreno (2007)* and extension to black water pixels in the Short Wave InfraRed (SWIR) (3) adjacency correction and (4) atmospheric correction using pre-calculated MODTRAN 5 Look Up Tables based on a rural aerosol model (*De Keukelaere et al., 2018*). Above water the SIMilarity Environment Correction (SIMEC) is used, which is based on the correspondence with the Near InfraRed (NIR) similarity spectrum and is described in *Sterckx et al. (2015)* and *Sterckx, Knaeps & Ruddick (2011)*. Above land, fixed background ranges are used.

A Rayleigh atmospheric correction algorithm, originally designed for MERIS, is also included in SNAP software. The current version of SNAP software also supports Sentinel-3

OLCI and Sentinel-2 MultiSpectral Instrument (MSI). Specifically, Rayleigh correction can be applied to: MERIS bands 1 to 15 (N1 format or MERIS 4th reprocessing format), Sentinel-3 OLCI L1B bands 1-21 and Sentinel-2 MSI L1C bands 1 to 9. The Rayleigh correction processor as it is described in S3TBX - Rayleigh Correction Tutorial (*Ruescas & Müller, 2021*) has five different outputs: Rayleigh optical thickness (ROT), Bottom of Rayleigh Reflectance (BRR), gaseous corrected TOA reflectance, TOA reflectance bands and air mass.

### Aim of the study

The aim of this study was dual: (1) to evaluate the possible differences between a partial atmospheric correction method accounting for Rayleigh scattering and a full atmospheric correction method (iCOR), applied on Sentinel-3 OLCI images for the study of shallow eutrophic lakes. Even though a full atmospheric correction method may be more suitable compared to Rayleigh correction for the calculation of water-leaving reflectance, possible large differences in processing time between the two methods, may be especially critical in cases of studies involving large time-series datasets. (2) to perform the atmospheric correction processing with free software (ESA's SNAP) in order to enhance the applicability of the methods. Even though several atmospheric correction software packages exist, many of them require programming language skills, or function as commercial software plugins, limiting their usability by end users.

## MATERIALS AND METHODS

### Study site

The study area is the Karla Reservoir, Thessaly, Greece (39°29′27″N 22°49′19″E, Fig. 1). Karla was a natural lake, which was drained in 1962. However, a series of negative consequences resulting from its drainage has led to its reconstruction in 2010, in an attempt to alleviate the negative effects (*Laspidou et al., 2017*). In its current state, the reservoir occupies a surface of 34 km$^2$ with a maximum water depth of 2 m (*Falaras, Koilakou & Tsoukalas, 2020*; *Papadimitriou et al., 2022*). As a shallow reservoir, Karla appears some important environmental implications, such as eutrophication and frequent and prolonged cyanobacterial blooms that produce toxins (*Gkelis et al., 2017*; *Oikonomou et al., 2012*; *Papadimitriou et al., 2022*). Occasionally, the severity of such blooms has been associated to mass kills of fish (*Papadimitriou et al., 2013*) and migrating birds (*Papadimitriou et al., 2018*).

### Satellite images

The satellite images, which were used in this study, were downloaded from the Copernicus Open Access Hub (https://scihub.copernicus.eu/), which provides full and free access to images of the Sentinel-1, Sentinel-2, Sentinel-3 and Sentinel-5P satellites. 53 cloud free, full resolution (300 m pixel size) Level-1B Sentinel-3 OLCI images (OL_1__EFR__ products), providing radiometrically calibrated, ortho-geolocated and spatially re-gridded Top Of Atmosphere (TOA) radiances, were used.
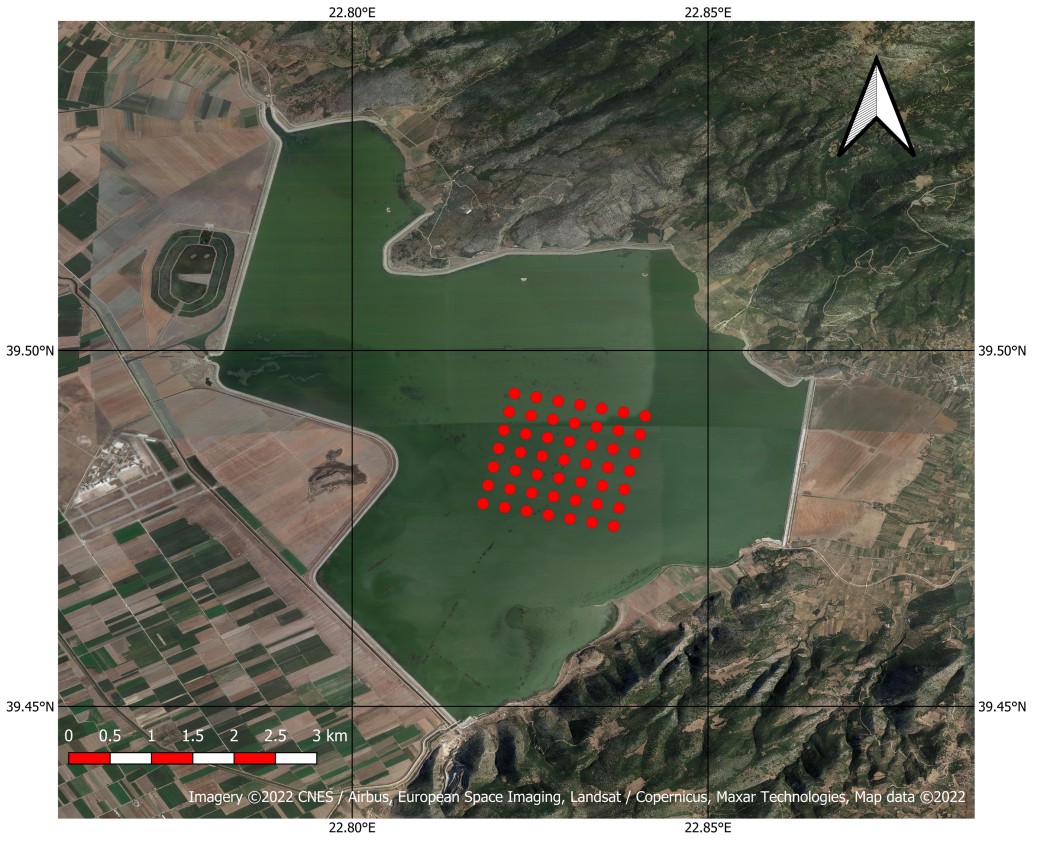

**Figure 1** Google Satellite image of the Karla Reservoir with the centres of the 49 pixels used in the study indicated with red dots (Imagery©2022 CNES/Airbus, European Space Imaging, Landsat/Copernicus, Maxar Technologies, Map data ©2022).

## Atmospheric correction application

Data from all images were converted from TOA radiances to Bottom Of Atmosphere (BOA) reflectance using both a partial atmospheric correction method accounting for Rayleigh scattering and a full atmospheric correction method (iCOR) and the results obtained with the two methods were intercompared (Fig. 2). Image processing was performed with SNAP-ESA Sentinel Application Platform v8.0 (http://step.esa.int) free software, which includes a Rayleigh correction algorithm, while iCOR is available as a SNAP plugin. Rayleigh correction is a straightforward procedure in SNAP, completing rather fast, *i.e.*, in approximately 5 min for a Sentinel-3 image. On the other hand, for the iCOR correction, several options have to be selected by the user. In this study, the Atmospheric Optical Transmittance (AOT), water vapor and ozone concentrations were estimated from the corresponding data included in the Sentinel-3 images *per ce*. In contrast to the Rayleigh correction, the iCOR procedure is rather time-consuming, taking about 2 h to be completed

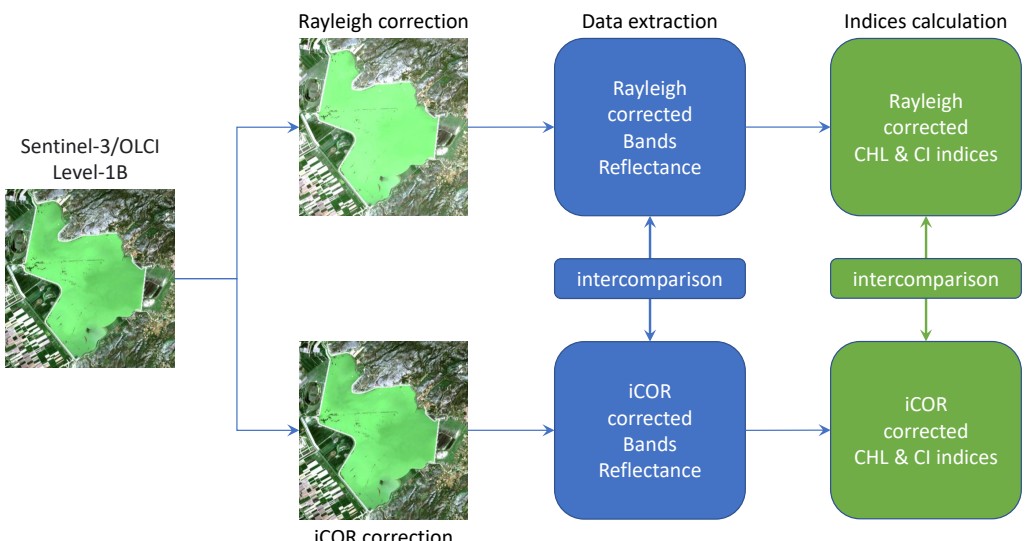

**Figure 2  Flowchart of the image processing with the two atmospheric correction methods (Rayleigh, iCOR).**

in a personal computer with a 7th Generation Intel® Core™ i5 4 core/4 threads processor and 16 GB of RAM.

From the 21 spectral bands (400–1020 nm) that exist in the Level-1B Sentinel-3 images and the Rayleigh corrected ones, 16 bands are included in the iCOR corrected images, since bands 13 (761.25 nm), 14 (764.375 nm), 15 (767.5 nm), 19 (900 nm) and 20 (940 nm) are used during the correction processing (Fig. 3). Accordingly, band intercomparison between Rayleigh and iCOR corrected images was performed for the bands common in both processes.

From all atmospherically corrected images with both methods, data were extracted for 49 pixels (7×7 rectangle) located approximately in the center of the Karla Reservoir (Fig. 1).

## Indices calculation

One of the most common variables used in lake monitoring through remote sensing techniques is the phytoplankton abundance. Accordingly, to further evaluate the above-described correction methods, two indices, *i.e.*, for chlorophyll (CHL) and phycocyanin (CI), were calculated from the atmospherically corrected spectral data from both methods, according to the following formulas, which provide accurate estimations in eutrophic waters (*Gitelson et al., 2008*; *Wynne et al., 2008*):

- $CHL = \left( \frac{1}{r_{665}} - \frac{1}{r_{708}} \right) \times r_{753}$
- $CI = -(r_{681} - r_{665} - (r_{708} - r_{665}) \times (\lambda_{681} - \lambda_{665})/(\lambda_{708} - \lambda_{665}))$

where r is the spectral reflectance at the indicated wavelength and $\lambda$ is the wavelength.

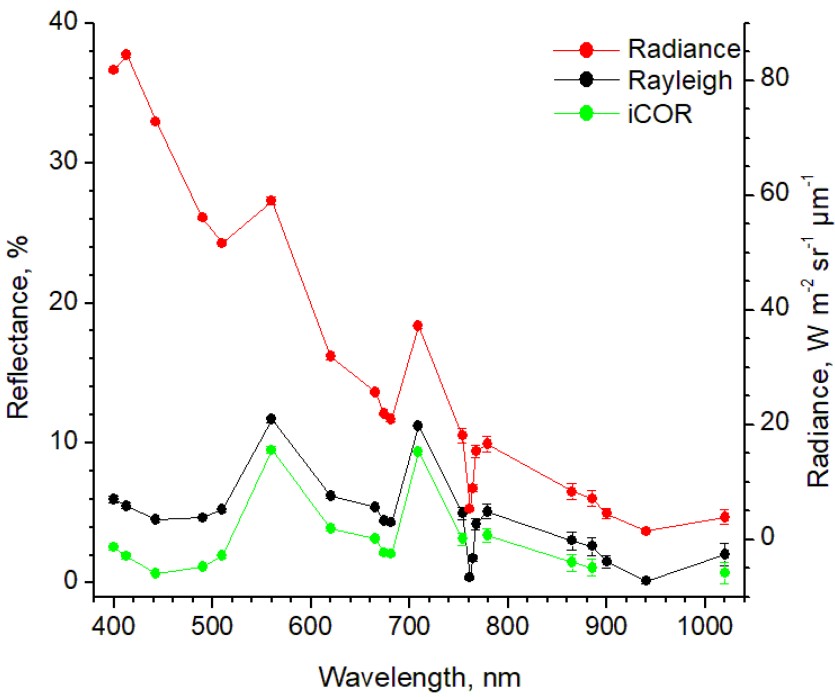

**Figure 3** **Band data from the original Level-1B image (Radiance) and the corresponding Rayleigh and iCOR corrected images (Reflectance) for 16/09/2017.** Data are average ± SD from 49 pixels.

## Statistics

For the comparison of the two correction methods a correlation analysis was performed using the JASP v. 0.14 software (JASP Team (2021). JASP (v. 0.14)). The Pearson's correlation coefficient (r), significance level (P) and intercept and slopes of their linear relationships are given. Comparisons were made for all 53 images both at pixel level and at pixel average level for each image/date (49 pixels per image). Correlations for each common band between the two correction methods (16 bands) and for chlorophyll (CHL) and phycocyanin (CI) indices were performed.

## RESULTS

### Band comparison

Band intercomparison revealed good correlations both at pixel level and pixel average level (Table 1), with correlation coefficients higher than 0.73. Especially for the bands used in chlorophyll and phycocyanin indices calculation, correlation coefficients higher than 0.89 were obtained (Fig. 4). As expected, for all bands Rayleigh corrected data show higher values compared to iCOR, since the former method concerns a partial atmospheric correction. However, the differences between the two methods are rather small, confirming that the Rayleigh correction accounts for the most dominant component of the TOA signal (*Shanmugam, Shanmugam & He, 2019*).
**Table 1  Band intercomparison statistics (intercept, slope and correlation coefficient *r*) between iCOR and Rayleigh corrected data.** For all bands $P < 0.0001$ and $N = 2597$.

| Band | Wavelength, nm | Intercept | Slope | r |
|------|----------------|-----------|-------|-----|
| 1 | 400 | 0.0410 | 0.6559 | 0.734 |
| 2 | 412.5 | 0.0431 | 0.6759 | 0.762 |
| 3 | 442.5 | 0.0379 | 0.6881 | 0.760 |
| 4 | 490 | 0.0326 | 0.7553 | 0.807 |
| 5 | 510 | 0.0288 | 0.8094 | 0.849 |
| 6 | 560 | 0.0162 | 0.9057 | 0.933 |
| 7 | 620 | 0.0180 | 0.8620 | 0.904 |
| 8 | 665 | 0.0194 | 0.8882 | 0.910 |
| 9 | 673.75 | 0.0199 | 0.8809 | 0.896 |
| 10 | 681.25 | 0.0198 | 0.8809 | 0.896 |
| 11 | 708.75 | 0.0170 | 0.8586 | 0.972 |
| 12 | 753.75 | 0.0172 | 0.9272 | 0.947 |
| 16 | 778.75 | 0.0162 | 0.9284 | 0.951 |
| 17 | 865 | 0.0144 | 0.9191 | 0.945 |
| 18 | 885 | 0.0140 | 0.9092 | 0.944 |
| 21 | 1020 | 0.0114 | 0.9122 | 0.952 |

## Indices comparison

High correlation patterns ($r > 0.95$) were also found for the chlorophyll and cyanobacteria indices, with the later performing slightly better (Figs. 5 and 6). As in the case of band intercomparison, there are differences in the indices' absolute values between the two methods, with the Rayleigh correction underestimating both indices, especially at high range values. However, the good correlations between the two methods, indicate that the Rayleigh corrected indices may be used alternatively for monitoring long term seasonal fluctuations of chlorophylls and phycocyanins and for accurate in-time detection of algal blooms.

In Fig. 7, maps for CHL and CI indices produced from Rayleigh corrected Sentinel-3 images are shown for two dates: June 1, 2018, a period of low pigments concentrations and July 21, 2019, a bloom period. Both indices are depicting well the spatial variability of the lake, providing the possibility to specify regions of special interest for monitoring or managerial purposes. For example, in the middle left parts of Figures a and b, high chlorophyll regions are apparent, due to their proximity to a channel which inputs—via a pumping station—already eutrophicated water with high pigment content.

## DISCUSSION

Atmospheric correction is a necessary processing step for the use of remote sensed data in ecosystem monitoring. Even though several correction software packages exist, many of them require programming language skills, or function as commercial software plugins. However, many end users would prefer free and simple to use software for the relevant image processing. Short processing time would be an additional advantage, in cases of

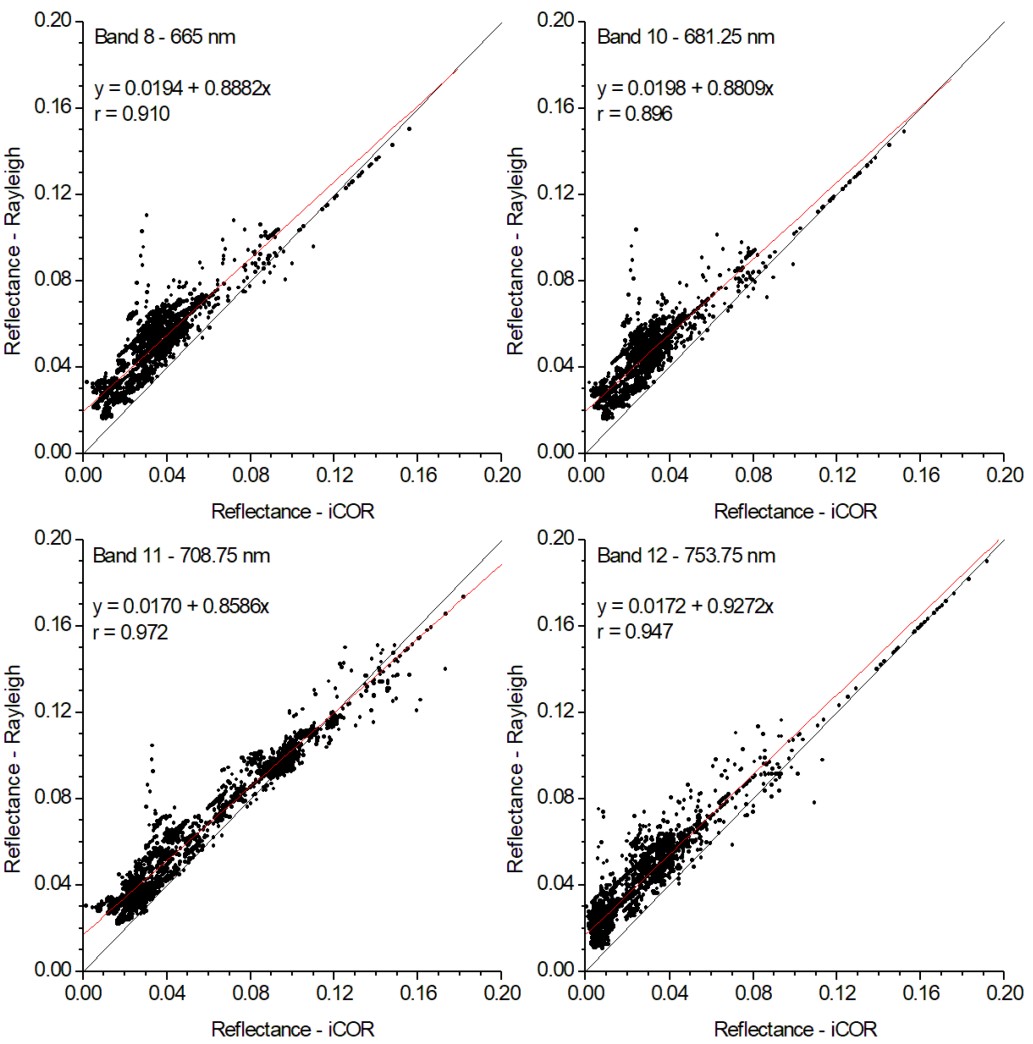

**Figure 4** **Correlations between the bands used in the calculation of chlorophyll and cyanobacteria indices.** For all bands $P < 0.0001$ and $N = 2597$.

large time-series ecosystem monitoring, whenever such data are available. The Sentinel-3 ESA satellites (and its predecessor MERIS) provide almost daily images globally since late 2016 and may be used for such purposes. Accordingly, in the present study the SNAP free software provided by ESA, was used to test two different atmospheric correction methods on Sentinel-3 OLCI images: a partial one accounting for Rayleigh scattering and a full one (iCOR) provided as a SNAP plugin.

iCOR has been systematically evaluated in comparison with several other full atmospheric correction methods (Acolite, C2RCC, l2gen, Polymer, Sen2Cor, ATCOR) for land images of Sentinel-2 and Sentinel-3 OLCI (*Rumora, Miler & Medak, 2020*; *Wolters et al., 2021*), for inland water images of Sentinel-2 MSI and Sentinel-3 OLCI (*Kravitz et al., 2020*; *Pahlevan et al., 2021*; *Pereira-Sandoval et al., 2019*; *Renosh et al., 2020*; *Warren et al., 2019*), even in arctic sea ice images of Sentinel-2 MSI (*König, Hieronymi & Oppelt, 2019*).
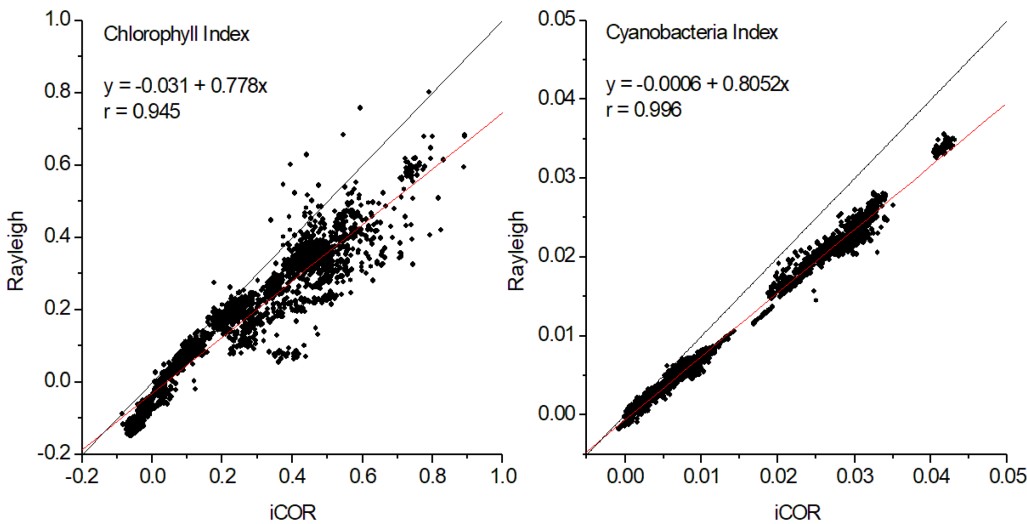

**Figure 5** **Correlations between iCOR and Rayleigh corrected data for chlorophyll and cyanobacteria indices at pixel level (2,597 pixels).** For both indices $P < 0.0001$ and $N = 53$.

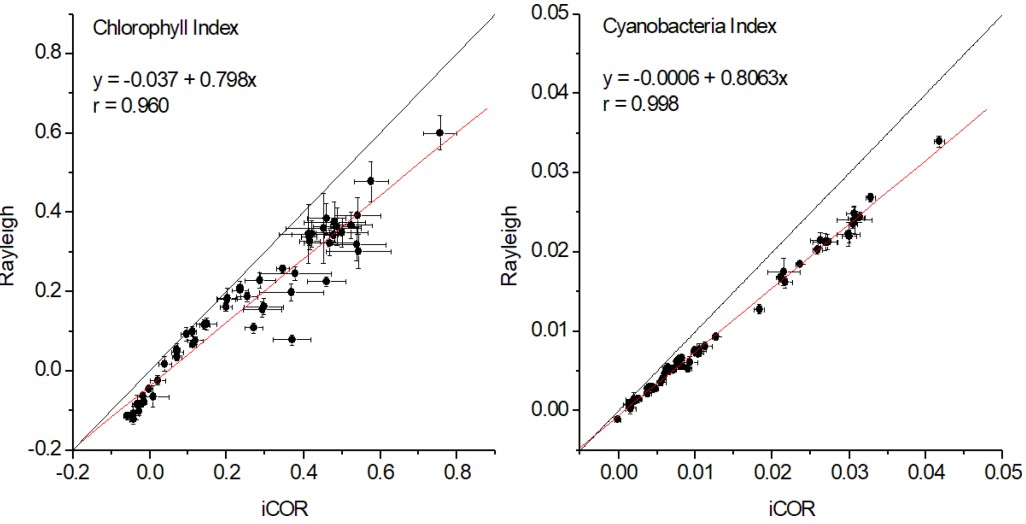

**Figure 6** **Correlations between iCOR and Rayleigh corrected data for chlorophyll and cyanobacteria indices at pixel average level (53 dates).** For both indices $P < 0.0001$ and $N = 53$.

Overall, it gives good results and is a reliable method for inland water images atmospheric correction.

The best practice for the validation of our results would be a direct comparison of the satellite derived CHL and CI indices with field measured chlorophyll and phycocyanin concentrations. However, this is a laborious and time-consuming task, which is beyond the aims of the current paper. Our strategy is to first compare which correction approach is more effective without losing relevant information and then focus on the comparison

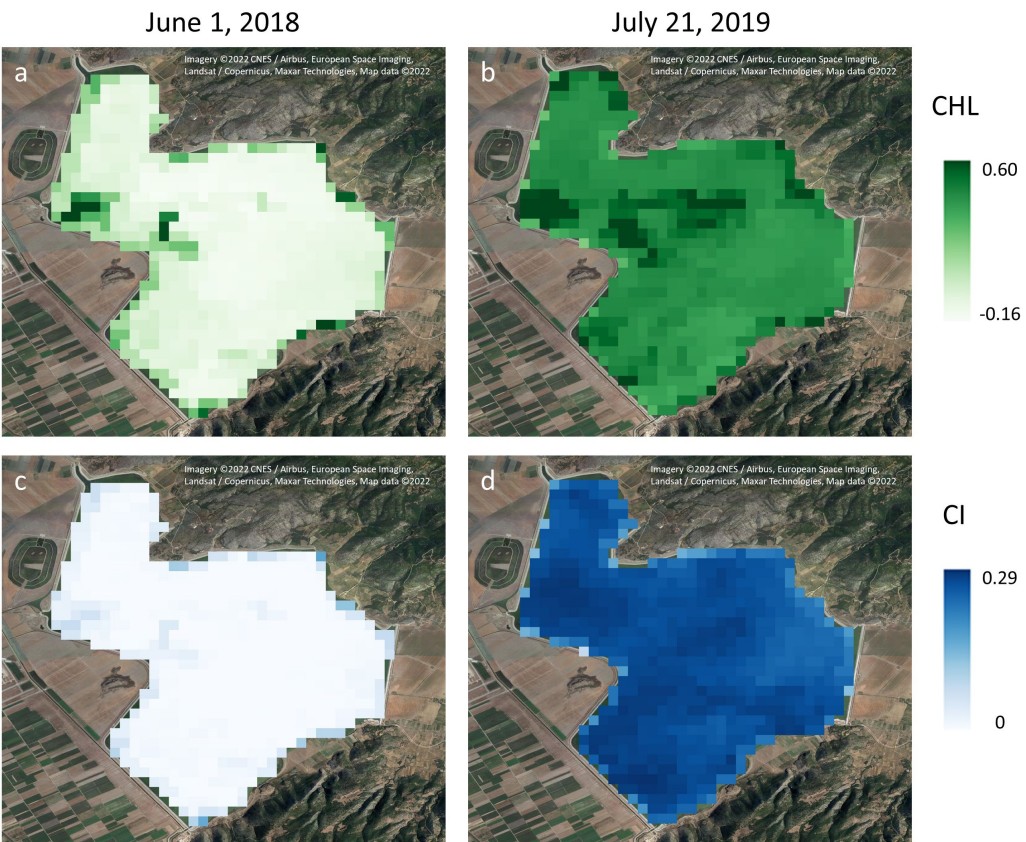

June 1, 2018      July 21, 2019

**Figure 7** **Chlorophyll (A, B) and phycocyanin (C, D) indices maps produced from Rayleigh corrected images for a low (A, C: June 1, 2018) and a high (B, D: July 21, 2019) pigment concentration date.** Chlorophyll (A, B) and phycocyanin (C, D) indices maps produced from Rayleigh corrected images for a low (A, C: June 1, 2018) and a high (B, D: July 21, 2019) pigment concentration date. A Google Satellite image is shown in the background (Imagery ©2022 CNES / Airbus, European Space Imaging, Landsat / Copernicus, Maxar Technologies, Map data©2022).

with field data. Such a task would require a very specific sampling strategy covering at least 1 calendar year, including episodic events, several sampling stations covering the spatial differences in the reservoir etc. Accordingly, since iCOR has already been evaluated against other correction algorithms and is considered a reliable atmospheric correction method, as mentioned above, our comparison of Rayleigh to iCOR may be considered as an indirect evaluation, which remains to be validated by combining *in situ* data.

In the above-described framework, the comparison of the two atmospheric correction methods in this study revealed very good correlations for all bands and indices (CHL and CI). Considering these results, it appears that for Sentinel-3 OLCI images in a shallow eutrophic reservoir such as Karla Reservoir, both methods can be used to calculate the CHL and CI indices with similar success. Therefore, one can use either of the two atmospheric corrections for the seasonal monitoring of the reservoir, without though the absolute values coinciding between the two methods. However, since the iCOR correction is much more demanding in terms of computational power and image processing time, it seems that the

partial correction of the Rayleigh method may be used alternatively, with obvious benefits in time and resource use efficiency, especially in cases of long time-series data.

Similar results have been reported by *Matthews, Bernard & Robertson (2012)* and *Matthews & Odermatt (2015)*, examining an algorithm for the calculation of chlorophyll-a in MERIS inland water images and stating that for broad trophic status assessment, simple Rayleigh atmospheric corrections are likely sufficient and avoid the more complicated and error-prone aerosol atmospheric corrections in turbid case II waters (waters which cannot be described by only one optical constituent of the water column). To the best of our knowledge, there has been no other research comparing a complete atmospheric correction with a partial atmospheric correction for Rayleigh scattering in a shallow eutrophic reservoir.

## CONCLUSIONS

The ESA's SNAP software used in this study, provides a free and user-friendly alternative for atmospheric correction of satellite images. Among others, it provides two correction methods, a partial one accounting for Rayleigh scattering and a full one (iCOR). The comparison of these two methods for Sentinel-3 OLCI images showed very good correlations for all bands ($r > 0.73$) and CHL and CI indices ($r > 0.95$). However, the 25 times faster and/or less resource demanding image processing of the Rayleigh correction method compared to iCOR may be of critical importance, especially in cases of long timeseries for monitoring algal blooms and water quality characteristics in shallow reservoirs. Even though it is not recommended to replace the full atmospheric correction algorithms, the application of only a partial correction for Rayleigh scattering in a shallow eutrophic reservoir seems sufficiently functional, with obvious advantages from time and resource use perspective. Additional research is needed to confirm our results in other shallow eutrophic lakes and probably examine and extend the applicability of the Rayleigh correction in general. To that purpose, comparisons with *in-situ* data for a full assessment of the prospects of applying only a partial atmospheric correction for Rayleigh scattering should be addressed.

### Funding
The authors received no funding for this work.

### Competing Interests
Konstantinos A. Kormas is an Academic Editor for PeerJ.

### Author Contributions
- Stefanos Katsoulis-Dimitriou performed the experiments, analyzed the data, prepared figures and/or tables, authored or reviewed drafts of the article, and approved the final draft.

- Marios Lefkaditis performed the experiments, analyzed the data, prepared figures and/or tables, authored or reviewed drafts of the article, and approved the final draft.
- Sotirios Barmpagiannakos performed the experiments, analyzed the data, prepared figures and/or tables, authored or reviewed drafts of the article, and approved the final draft.
- Konstantinos A. Kormas conceived and designed the experiments, analyzed the data, authored or reviewed drafts of the article, and approved the final draft.
- Aris Kyparissis conceived and designed the experiments, performed the experiments, analyzed the data, prepared figures and/or tables, authored or reviewed drafts of the article, and approved the final draft.

### Data Availability

The 53 Sentinel-3 images used in the study are publicly available from European Space Agency (https://scihub.copernicus.eu/).

To find the Sentinel-3 images, use the following search parameters: Area of Interest Karla lake, Level-1B Sentinel-3 OLCI images (OL_1__EFR__ products).

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
