# Peer review of "Comparison of iCOR and Rayleigh atmospheric correction methods on Sentinel-3 OLCI images for a shallow eutrophic reservoir"

_PeerJ, doi:10.7717/peerj.14311_

## Round 0.1 · original submission · Major Revisions

Dear Authors,
Thank you very much for submitting your work to PeerJ. As you shall notice, the reviewers have commented on your manuscript and are suggesting a major revision. Kindly go through their comments and revise your manuscript accordingly.

Thank you once again.

Best regards
Gowhar

·

Basic reporting

The manuscript is written clearly, with excellent grammar and spelling. The literature cited is relevant and sufficient with no major omissions. The article is structured appropriately, with adequate and relevant figures and tables. The raw data is available from ESA as stated. The hypothesis is clear, concise and adequately supported by the materials.

Experimental design

The article is within the aims and scope of the journal. The research question is relevant and well-defined. The methodology is rigorous and repeatable.

Validity of the findings

The results are valid and meaningful, and support and build on other similar findings in literature. Underlying data have been provided via ESA. Conclusions are valid and succinct.

Additional comments

A PDF is attached with further comments. Some minor corrections are required before publication.

Reviewer 2 ·

Basic reporting

The research article submitted entitled Comparison of iCOR and Rayleigh atmospheric correction methods on Sentinel-3/OLCI images for a shallow eutrophic reservoir meets scientific content, innovations, constructive methodology, well-written results, and conclusions. The research topic covered is unique and innovative. The current research adds knowledge, methods, and findings to readers, scientific content, and modeling, are unique.

Abstract: The authors must revise it, as to how this study can be a role model for further investigation. Highlight major results with statistics. Rest ok

Introduction and aim of the study: Must be rewritten, as in current form lacks novelty, innovations, hypotheses, recent studies on the current topic, and various latest techniques used are missing.
Study area: Include more relevant literature for the introduction of your study area to the unknown authors for greater readability.

Methods: In the current form methodology flowchart is missing, the authors must include it with a detailed discussion on how they followed the processes to obtain the results.

Results: The results are too short. No significant knowledge to the readers and already known facts are discussed. Authors also need to elaborate discussion as in its current form is not acceptable. It is requested to revise the entire content and refer to the format of other research articles published in PeerJ. It’s also requested to authors keep all the maps in the same template, shape, size, and figures, as all the maps in current research are of different shapes and sizes. The font size, quality, template, and resolutions must be high and clear.

Conclusion: Again it is too short and no significant outcome of research missing. Please add major highlights and future scope and way forward from current research.

Experimental design

no comment

Validity of the findings

no comment

Additional comments

no comment

Reviewer 3 ·

Basic reporting

This study has tried to develop an algorithm that could estimate the algal blooms in the eutrophic waters using the Sentinel-3/OLCI images. The method has used iCOR correction image processing and Rayleigh correction for accounting for the atmospheric corrections. Overall the work has been well reported.

Experimental design

Though, the experimental design is well orchestrated. However I was perplexed to see, has there been validation with the observed data? This needs to be taken care of

Validity of the findings

Validation needs to be established with the observed data for both the atmospheric correction methods

Additional comments

All the sections are not written with enough literature review. This paper at present seems to be a technical note and not a full-fledged research article. It needs a significant increase in reporting and discussion in each section.

---

## Round 0.2 · Minor Revisions

Dear Authors,
As you shall see that the reviewers have now commented on your manuscript and are suggesting some further revisions. Kindly incorporate them.

Moreover, as suggested by one of the reviewers that it is important to highlight why observational data was not used for validation in the manuscript. Therefore, it is strongly recommended to incorporate a subsection within the discussion section, wherein this issue needs to be discussed.

Rest please take note of other suggestions as well.


Best regards
Gowhar Meraj

Reviewer 2 ·

Basic reporting

The present study was carried out to evaluate the possible differences between a partial atmospheric correction method, accounting for Rayleigh scattering, and a full atmospheric correction method (iCOR), applied to Sentinel-3 OLCI images of a shallow, highly eutrophic water reservoir. For the complete evaluation of the two methods, in addition to the comparison of the band reflectance values, chlorophyll (CHL) and cyanobacteria (CI) indices were also calculated and their values were intercompared.

The study is relevant to the readers of environmental sciences, geology, geography, remote sensing, GIS, geomatics, geoinformatics, atmospheric sciences, etc. The article is well written and structured, with sufficient scientific understanding. The hypothesis is well-framed and justified. The figures and tables are well represented. The literature review used for the formulation of the hypothesis is sufficient for greater clarity and background.

A few suggestions which can improve the quality of the manuscript:
1. The English language needs minor corrections.
2. The quality of figures can be improved.
3. Figure 1 can be replaced by Sentinel-3 OLCI images as Google Earth images may be restricted for use in any scientific investigations as the authors already have high resolution images.

Experimental design

The aims and objectives are within the scope of the journal. The research gaps have been identified and addressed correctly for meaningful knowledge. The results are well drafted and analyzed using the latest indices using scientific methods and innovative. The article is expected to receive a good number of citations in the coming days.

A few suggestions which can improve the quality of the manuscript:
1. The images used in Figure 2 are too small, the image size can be improved.
2. The two indices, i.e., for chlorophyll (CHL) and phycocyanin (CI), resultant products are missing and must be displayed.
3. The result in its current form is too short and must be elaborated in the context of the study.

Validity of the findings

The study is innovative and provides a robust, statistically sound, and controlled environment. The conclusion is well written, linked to research questions and supporting the findings and results.

A few suggestions which can improve the quality of the manuscript:
1. The discussion is too short and must be elaborated upon in the context of the study.

Reviewer 3 ·

Basic reporting

The work has been reported in clear and concise manner. Lierature has been cited well. Figures and tables have been professionally structured.

Experimental design

The study design is robust. The investigation has been conducted scientifically robust manner. Methods have been described sufficiently.

Validity of the findings

It is suggested that the arguments provided by the authors regarding the use of observational data for validity assessment of algorithms must be included in the manuscript as well in the discussion section. This is very important for the readers to know, why field data has been used for validation.

Additional comments

I recommend minor revision

---

## Round 0.3 · accepted · Accept

Thank you very much for incorporating all the suggestions made by the reviewers in each round of review.